# Structural Relationship of Causes and Effects of Construction Changes: Case of UAE Construction

**Ismail Abdul Rahman [1], Abdulla Eisaa Saleh Al Ameri [2], Aftab Hameed Memon [3,*], Nashwan Al-Emad [1] and Ahmed S. A. Marey Alhammadi [2]**

[1] Faculty of Civil and Environmental Engineering, Universiti Tun Hussein Onn Malaysia (UTHM), Batu Pahat 86400, Johor, Malaysia; ismailar@uthm.edu.my (I.A.R.); df080116@gmail.com (N.A.-E.)

[2] Faculty of Business and Technology Management, Universiti Tun Hussein Onn Malaysia (UTHM), Batu Pahat 86400, Johor, Malaysia; alaamriab@hotmail.com (A.E.S.A.A.); gp170093@siswa.uthm.edu.my (A.S.A.M.A.)

[3] Department of Civil Engineering, Quaid-e-Awam University of Engineering, Science and Technology, Sindh 67450, Pakistan

*   Correspondence: aftabm78@hotmail.com

**Abstract:** Changes during construction is one of the critical issues faced in the construction industry. Effective management of construction changes will reduce the financial burden faced in construction projects due to cost overrun, and practitioners will be able to complete projects on time. On the other hand, construction changes exert severe effects on project performance. Hence, this paper uncovers several changes occurring in construction projects. It also evaluates the effect on various parameters of project performance due to changes. This was done by uncovering the underlying causes and effects of changes through the PLS method of structural equation modeling technique. SmartPLS software was used to develop and evaluate the study model based on 58 change causes and 48 change effects that were identified from the literature review. Causes of changes were categorized into three constructs which are client-related causes (CLE), consultant-related causes (COS), and contractor-related causes (CON). At the same time, the effects variables were grouped as Time Overrun (TO), Cost Overrun (CO), and Quality (QA). The survey data for generating the model was collected from 218 practitioners working on construction megaprojects of the UAE. Assessment on the constructed model found that the contractor (CON) group is the most influential group of causes with the highest values of the model's predictive explanatory power (accuracy), which is 0.396, 0.339, and 0.410 to time overrun, cost overrun, and quality assurance of the effects groups, respectively. At the same time, the Quality Assurance (QA) group is considered the most substantial parameter which are affected due to changes occurring in construction projects of UAE. This model is deemed beneficial for the UAE construction industry in facilitating the effective recognition of possible causes and effects of change among the UAE construction projects. As a result, the practitioners will make necessary arrangements to control the potential changes in future projects.

**Keywords:** PLS-SEM; changes in construction; causes and effects of changes; UAE construction

## 1. Introduction

In UAE, the construction is considered the leading sector that drives the country's economy [1]. It is an essential element in driving the UAE economy to a better standard and continues to be an essential task in the UAE's development, urbanization, and industrialization [2,3]. It has helped to translate the vision 2025 of UAE into a more innovative, planned development status [4]. It plays an essential role in the creation of jobs. It employs over 2,000,000 workers; accounting for about 7.2 percent of the total workforce, stabilizing the economy [5]. UAE's construction industry has expanded rapidly. A number of megaprojects such as Dubai Creek Harbour, Al-Maktoum International Airport, and Dubai, the sustainable city (net-zero energy city in Dubai), have given exposure to the construction industry in the world.

Unfortunately, construction works in UAE experience changes during work in progress, which significantly affects progress and performance due to uncertainty. The changes have a severe impact on project cost, schedule, and quality. It is reported that the Dubai metro project faced a delay of five years to complete due to changes occurring through the project lifecycle. As a result, the cost was increased by 85%, as cited in [6]. Although every stakeholder involved in construction works aims to achieve project completion within planned scope, time and budget can become challenging when changes occur in project activities [7]. Changes in the project are unavoidable [8] and create serious adverse effects [9]. Hence, it is very imperative to control changes in construction. This can be effectively performed by identifying critical causes of changes. Thus, this study aims to uncover underlying causes and effects caused by changes in construction projects of UAE through advanced multivariate techniques of Structural Equation Modeling (SEM). Identification of the causes of changes and relative effects are helpful in planning the strategies for executing the works successfully.

## 2. Causes and Effects of Changes in Construction Projects

The construction industry is considered to be the primary contributor to developing socio-economic status. Unfortunately, construction projects often face common issues such as time overrun, waste generation, poor productivity, etc. These issues are due to several reasons. One of the reasons is that the construction projects are uncertain. Due to the uncertainty phenomenon in construction works, most construction projects experience changes during the execution of the work. Change in the construction industry usually arises when the scope of work performed differs from the scope of work outlined in the contract documents [10]. Adding or reducing the scope or even modifying contract conditions affect construction cost, time, and quality [11].

Changes in construction are inevitable which are notified by change orders [12]. These are considered a serious concern of the practitioners as they can lead to project failure, time overrun, cost overrun, and inadequate project quality [13]. A study of Saudi Arabian construction works showed that 70% of the project faced extended project time. These changes occur due to variations in design, expenses delays, inadequate planning and scheduling, absence of site management by contractors, labor deficiency, and financial problems with a project's contractors [14]. Ahmed et al. [15] reported that the construction projects in Pakistan are affected by the huge claims of cost escalation. One of the primary reasons of this escalation is the extra time required for the project due to changes. Changes cause numerous negative impacts on project performance [16] and may lead to disputes and declines in productivity [11,17].

Change management is amongst the major challenges which hurdle the success of the projects, and it may be caused by design error, mistakes in drawings, amendment in contract condition, modification in scope, etc. A research study revealed that construction projects face 10–17% of the overrun in project cost due to changes [17]. Change can be difficult for all stakeholders involved in project management and can lead to contractual disputes [13]. It can create misunderstanding between parties such as contractors who consider consultants and clients responsible for any change.

The owner may perceive losses as a result of change caused by the contractor's poor management [18]. Today, managing changes is a critical issue that can have a significant impact on project performance if not handled properly [19]. Sun and Meng [20] reported that conditions are the major source of change in any project. Generally, any event which causes alteration or modification of the work item is considered to change. Design and scope changes are the most common reasons for changes in any project [21], but unfortunately, only a few studies have paid proper attention to this issue [22].

Similarly, lack of contractor's experience in similar projects or poor financial condition of stakeholders also causes changes in projects [13]. Alaryan and Dawood [23] pointed out that changes in design or material specification and inconsistency in contract documents also cause change. In addition, financial difficulties of the contractor, slow decision making,

or inflexible nature of the owner also cause changes in projects [24]. A comprehensive review of the literature identified 53 common causes of construction changes and 48 effects occurring due to changes. A broad examination of the factors highlighted that the causes could be classified based on the source or controlling stakeholder. Hence, the factors were categorized in the three groups: client-related factors, consultant-related factors, and contractor-related factors. The effects were related to the basic parameters of projects performance as time, cost, and quality. The effects of changes were classified into three categories: time over, cost overrun, and quality assurance. The list of the causes and effects of the changes was tabulated and presented in Table 1.

**Table 1.** Causes and effects of changes constructing constructs.

| S. No | Construct | Factor Code | Description |
|---|---|---|---|
| | | **CAUSES OF CHANGES** | |
| 1 | | CLE01 | Clients' financial problems |
| 2 | | CLE02 | Late payments |
| 3 | | CLE03 | Delay in order issuance by clients |
| 4 | | CLE04 | Owners' needs |
| 5 | | CLE05 | Economic inflation |
| 6 | | CLE06 | Elections and clients' representative changes |
| 7 | | CLE07 | Inadequate understanding of clients' needs |
| 8 | | CLE08 | Conflicts with consultant and contractor |
| 9 | CLE | CLE09 | Multiple contractors |
| 10 | (Client-Related Factors) | CLE10 | Clients' organizational problems |
| 11 | | CLE11 | Unprofessional clients |
| 12 | | CLE12 | Clients' authority change |
| 13 | | CLE13 | Inadequate site mobilization by contractor |
| 14 | | CLE14 | Inadequate bidding documents by clients |
| 15 | | CLE15 | Lack of coordination |
| 16 | | CLE16 | Replacement of key personnel by clients |
| 17 | | CLE17 | Lack of capable clients representative |
| 18 | | CLE18 | Skill shortage on certain trades |
| 19 | | CLE19 | Unsafe practices during construction |
| 20 | | CST01 | Poor material specifications |
| 21 | | CST02 | Lack of scheduling and planning |
| 22 | | CST03 | Poor site and work investigation by consultant |
| 23 | | CST04 | Late revision of designs |
| 24 | | CST05 | Poor site management team |
| 25 | | CST06 | Inexperienced consultant |
| 26 | | CST07 | Poor estimations of cost and quantity |
| 27 | COS | CST08 | Multiple consultants |
| 28 | (Consultant-Related Factors) | CST09 | Poor investigation of project location |
| 29 | | CST10 | Poor consultant coordination |
| 30 | | CST11 | New regulations and codes |
| 31 | | CST12 | Poor prediction of equipment types |
| 32 | | CST13 | Site restrictions |
| 33 | | CST14 | Weather conditions |
| 34 | | CST15 | Geological problems |
| 35 | | CST16 | Poor distribution of labor |
| 36 | | CON01 | Inexperienced subcontractors |
| 37 | | CON02 | Subcontractors' financial problems |
| 38 | | CON03 | Errors in contractual documents |
| 39 | | CON04 | Problems with other organizations |
| 40 | | CON05 | Government pressure |
| 41 | | CON06 | Design errors |
| 42 | | CON07 | Large amount of labor costs |
| 43 | | CON08 | Conflicts with residents |
| 44 | CON | CON09 | Delay in providing utilities |
| 45 | (Contractor Related Factors) | CON10 | Owners' expectations and quality improvement by client |

**Table 1.** *Cont.*

| S. No | Construct | Factor Code | Description |
|---|---|---|---|
| 46 | | CON11 | Large amount of overhead costs (e.g., office rents, contract costs, etc.) |
| 47 | | CON12 | Unavailability of technical professionals in the contractor's organization |
| 48 | | CON13 | Lack of contractor's administrative personnel |
| 49 | | CON14 | Low level of labor efficiency/productivity |
| 50 | | CON15 | Inadequate skill of equipment-operator |
| 51 | | CON16 | Poor programming of material procurement |
| 52 | | CON17 | Non-familiarity of contractor with local regulations |
| 53 | | CON18 | Poor inspection and supervision by contractor |
| | **EFFECTS CAUSED DUE TO CHANGES** | | |
| 1 | | TO01 | Delay in completion schedule |
| 2 | | TO02 | Logistics delays |
| 3 | | TO03 | Slower project progress |
| 4 | | TO04 | Decrease in productivity |
| 5 | | TO05 | Delay completion schedule |
| 6 | | TO06 | Dispute between owner and contractor |
| 7 | TO | TO07 | Decrease in productivity of workers |
| 8 | (Time Overrun) | TO08 | Additional specialist personnel |
| 9 | | TO09 | Cost overruns due to inflation and fluctuations |
| 10 | | TO10 | Addition of work |
| 11 | | TO11 | Deletion of work |
| 12 | | TO12 | Rework/redesign |
| 13 | | TO13 | Work duration extension |
| 14 | | TO14 | Productivity degradation |
| 15 | | CO01 | Increase in overhead expenses |
| 16 | | CO02 | Increase the cost of the projects |
| 17 | | CO03 | Additional money for contractor |
| 18 | | CO04 | Delay in payment |
| 19 | | CO05 | Additional specialist equipment |
| 20 | | CO06 | Additional health and safety equipment/measure |
| 21 | | CO07 | Unnecessary procurement |
| 22 | CO | CO08 | Accumulations of interest rate on the capital to finance the project |
| 23 | (Cost Overrun) | CO09 | Waste on abandoned work |
| 24 | | CO10 | Demolition costs |
| 25 | | CO11 | Increase in overheads |
| 26 | | CO12 | Additional equipment and materials |
| 27 | | CO13 | Additional payment to contractors |
| 28 | | CO14 | Interrupted cash flow |
| 29 | | CO15 | Increased retention/contingency sum |
| 30 | | CO16 | Overtime costs |
| 31 | | CO17 | Litigation costs |
| 32 | | QA01 | Rejected material |
| 33 | | QA02 | Poor quality of materials |
| 34 | | QA03 | Changes in materials specifications |
| 35 | | QA04 | Problems with new materials |
| 36 | | QA05 | Changes in material types and specifications during construction |
| 37 | | QA06 | Replacement/substitution of materials |
| 38 | | QA07 | Quality degradation |
| 39 | QA | QA08 | Damage to reputation |
| 40 | (Quality Assurance) | QA09 | Degradation of health and safety |
| 41 | | QA10 | Demolition and re-work |
| 42 | | QA11 | Decrease in quality of work |
| 43 | | QA12 | Complaints of one or more of the parties to the contact |
| 44 | | QA13 | Rework of bad quality performance |
| 45 | | QA14 | Slow response and poor inspection |
| 46 | | QA15 | Extension of time on the project |
| 47 | | QA16 | Wastage and under-utilization of man-power resources |
| 48 | | QA17 | Abandonment of building project |

## 3. Structural Equation Modeling

Structural Equation Modeling (SEM) is a method for explaining the relationship between many variables. The variables that can be measured directly are known as observed variables. In contrast, the variables that cannot be measured directly (latent variables) are dependent on the factors that are associated with them. A typical SEM framework includes a structural model that connects latent variables and a measurement model that links latent variables to observed variables. SEM is particularly popular in the social sciences and psychology, where unmeasured quantities and psychological constructs like human intelligence and creativity can be related to and investigated using observed data [25]. SEM is a technique that can be used for both confirmation and exploration. In SEM, each path model comprises two submodels: a structural or inner model and measurement or outer model. The structural model determines the relationship between latent variables. The measurement model determines the relationships between latent variables and their manifests [26]. This powerful method can also be used to analyze models with poorly measured variables, and many research issues in construction management and engineering can be addressed [27]. SEM has demonstrated numerous advantages in prediction and theory development. Even with small sample size, SEM is considered a very effective technique for assessing the reliability of multi-item construct measures [28].

Being a robust analysis technique, the use of SEM is proliferating [29]. SEM is applicable for decision support system development, predictive models, risk analysis, and so on. For example, Doloi et al. [30] used SEM to quantify the relationships between the causes of project delays in Indian construction projects. Islam and Faniran [31] drew attention to the SEM's effectiveness in determining relationships between multiple independent variables. Memon and Rahman [32] used SEM to identify cost overrun impediments in Malaysia. Khahro et al. [33] modeled the factor of green procurement using a PLS approach to SEM.

In Structural Equation Modelling, two approaches are commonly used [34], namely, Structural Equation Modeling Using Covariance (CB-SEM) and Variance-Based Structural Equation Modeling (VB-SEM), also known as Partial Least Square Structural Equation Modeling (PLS-SEM). Partial Least Square Structural Equation Modeling (PLS-SEM) is also known in the literature as component-based SEM or PLS path modelling. In contrast to CB-SEM's goal of obtaining a good fit, the PLS approach aims to get determinate values of the latent variables for predictive purposes [35]. The PLS-SEM method aims to maximize the explained variance of the dependent latent constructs [36]. As a result, the entire process is optimized for prediction rather than goodness-of-fit. PLS-SEM can be used for confirmatory analysis and for exploratory studies where the theoretical foundation is lacking. It is more robust than CB SEM, has fewer identification issues, and works well with small and large samples. Furthermore, PLS does not make any prior distributional assumptions.

## 4. Model Development

Partial Least Square Structural Equation Modeling (PLS-SEM) method was used for measurement analysis and hypotheses. In this study, PLS-SEM model was developed and run with the software package SmartPLS v3.0. Initially, a hypothetical model is developed that serves as a basis for testing the relationships among variables [37]. The hypothetical model has two latent variables known as exogenous and endogenous, which act as the PLS model's structural (inner) part. The exogenous variables are represented by three groups of change causes: client-related causes [CLE], consultant-related causes [CST], and contractor-related causes [CON]. Whereas, endogenous variables represent three groups of change effects: time overrun [TO], cost overrun [CO], and quality assurance [QA].

Based on the factors and constructs listed in Table 1 above, the model was constructed based on 53 causes of changes groups in three exogenous constructs and 48 change of effects categorized in three endogenous constructs. The model development aimed to assess the relationship between causes and effects. The endogenous and exogenous constructs are connected to form the structural component of the model [38]. A hypothetical model was

developed for assessing the relationship between the causes and effects of changes, as shown in Figure 1.

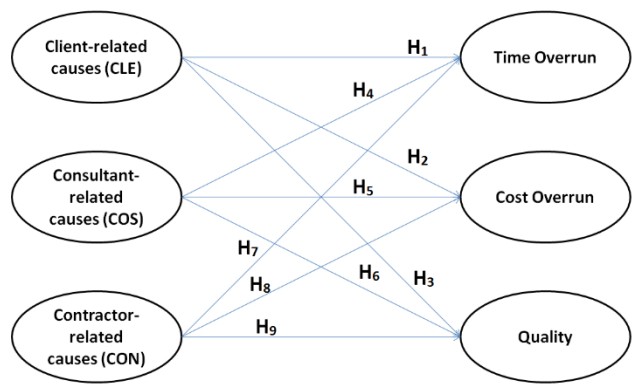

**Figure 1.** Hypothetical model.

In Figure 1, it is observed that there are 9 hypotheses developed to test the cause-and-effect relationship of changes as described below.

**Hypothesis 1 (H1).** *CLE has a significant relationship with TO.*

**Hypothesis 2 (H2).** *CLE has a significant relationship with CO.*

**Hypothesis 3 (H3).** *CLE has a significant relationship with QA.*

**Hypothesis 4 (H4).** *CST has a significant relationship with TO.*

**Hypothesis 5 (H5).** *CST has a significant relationship with CO.*

**Hypothesis 6 (H6).** *CST has a significant relationship with QA.*

**Hypothesis 7 (H7).** *CON has a significant relationship with TO.*

**Hypothesis 8 (H8).** *CON has a significant relationship with CO.*

**Hypothesis 9 (H9).** *CON has a significant relationship with QA.*

## 5. Data Collection

Data collection was carried out through a survey where the respondents were construction practitioners randomly selected based on their working experience in handling a number of mega projects in Dubai. The survey was conducted using Google online survey application. The respondents were asked to choose the degree of importance of 58 change causes affecting 48 change effects based on a 5-point Likert scale. In addition, the respondents were asked to indicate the level of significance for each cause and effect of change as 1 for not significant, 2 for slightly significant, 3 for moderately significant, 4 for very significant, and 5 for extremely significant. This survey resulted in acquiring 218 valid responses as demographic information in Figure 2.

Figure 2 indicates that most of the respondents (67%) have bachelor's degrees. In terms of experience, more than half of respondents (56.9%) had been working for less than five years in the UAE construction industry. In addition, it was observed that two-thirds of the respondents (71.6%) are technical workers, 15.1% of the respondents hold the position of executive management, and 13.3% of respondents are senior managers.

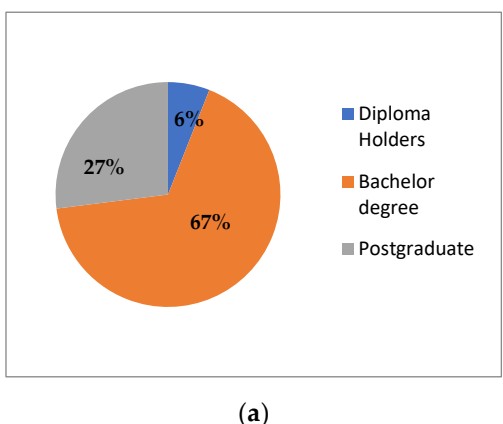
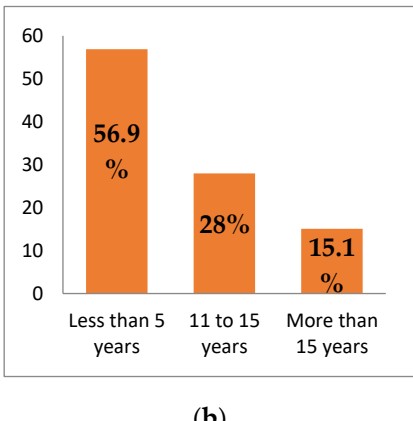

(**a**)                                     (**b**)

**Figure 2.** Demography of the respondents. (**a**) Academic qualification. (**b**) Working experience.

## 6. Model Evaluation

The two stages of model evaluation in SEM modeling are conducted for measurement (outer) and structural (inner) models. The assessment was conducted on the relationship between constructs and their related indicators for the measurement stage, while the structural evaluation was conducted between exogenous and endogenous constructs.

### 6.1. Evaluation of Measurement Model

Evaluation of measurement model is to check the internal consistency of the model and to evaluate whether relationships between independent and dependent variables are adequate or not [28]. For the reflective model approach, the evaluation of the measurement model is conducted in three stages. The first stage is to evaluate the model performance after each model computational iteration (individual item reliability) where indicator factor loading < 0.5. The second stage is to check the construct's convergent validity and reliability include composite reliability (CR > 0.708), average variance extracted (AVE > 0.5), and Cronbach's alpha ($\alpha \geq 0.7$). The third stage is confirming the discriminant validities of the model, where the square root of the average variance extracted (AVE) is more than correlation values between the other exogenous constructs. The process of model evaluation and iteration can be carried out alternatively until all evaluation criteria are fulfilled.

Evaluation of the measurement involved six iterations and assessment processes, resulting in a deletion of 70 indicators from the construct with low factor loading (<0.5). Deleting these indicators caused an improved value of the average variance extracted (AVE) to an acceptable level from 0.513 to 0.612. The final model showing the results of paths and strength of the factor are shown in Figure 3 where latent variable 1 represents the client-related causes, latent variable 2 represents a consultant-related group of causes, and latent variable 3 describes the contractor-related causes. Similarly, latent variable 4 illustrates time overrun, latent variable 5 represents cost overrun, and latent variable 6 represents quality-related factors.

Figure 3 shows the final constructed PLS model with path coefficients (β value) in the range of 0.119 to 0.485. This indicates that contractor (CON) has the highest β value of 0.485, showing the group has a strong relationship with the effects of change in the construction industry [28]. Research points to the contractor as the responsible party that carries most of the risks for the changes in the project. This is because sometimes the contractor has to perform work different from that required by the contract documents [10]. The overall model can be classified as having significant explaining power based on the coefficient of determination ($R^2$) value. Among the change effects, Quality Assurance (QA) is the significant group with the highest $R^2$ value. Regression analyzes the interaction effects [39] of exogenous and endogenous variables. Overall, it can be seen that the changes have a significant effect on time overrun, cost overrun, and quality. Shoar and Chileshe [22] also confirmed that the changes significantly affect time and cost overrun.

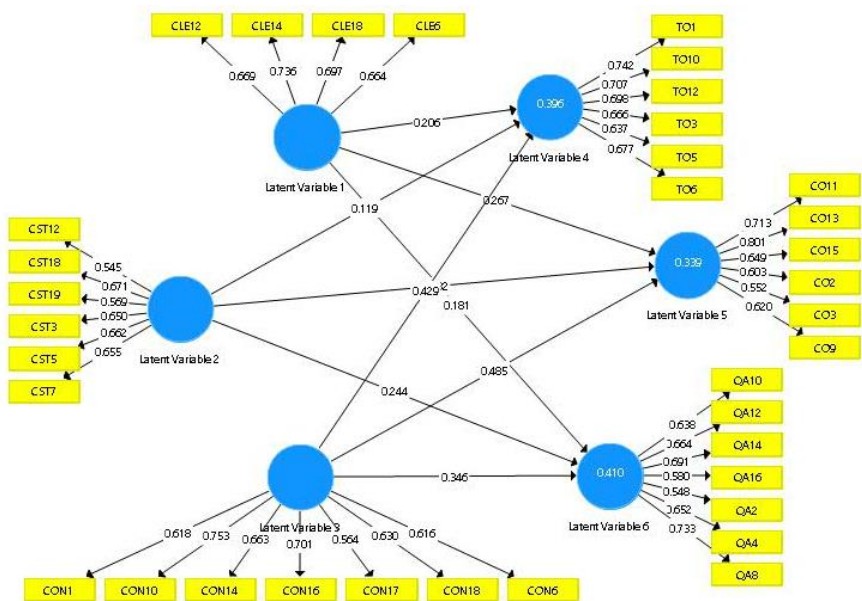

**Figure 3.** Final PLS Model of Causes and Effects of Changes in Construction Projects.

Furthermore, the developed model is categorized as reflective since the indicators' path is in an outward direction from the constructs. Hence, the model evaluation is based on nostalgic model specifications and criteria. The overall results of the measurement model evaluation are as presented in Table 2.

**Table 2.** Overall performance of the measurement model.

| No | Assessment | Achievement | | | |
|---|---|---|---|---|---|
| 1 | Individual item reliability | Outcome: 6 iterative processes were carried out and 70 weak factors were omitted leaving 36 significant manifests for the final output | | | |
| | | | Cronbach's alpha | rho_A | Composite reliability | Average variance extracted (AVE) |
| 2 | Convergent validity | CLE | 0.639 | 0.641 | 0.786 | 0.579 |
| | | CO | 0.655 | 0.664 | 0.783 | 0.521 |
| | | CON | 0.752 | 0.748 | 0.828 | 0.547 |
| | | CST | 0.754 | 0.762 | 0.835 | 0.503 |
| | | QA | 0.713 | 0.711 | 0.814 | 0.569 |
| | | TO | 0.767 | 0.775 | 0.833 | 0.518 |
| 3 | Discriminant validity-Cross-loading | Cross loading values of the model are presented in Appendix A. The results show that the cross-loading value for each manifest variable is higher in their relative latent variable than other latent variables (as indicated with bold font). This has confirmed the discriminant validity of the model. | | | | |
| | | | CLE | CST | CON | CO | QA | TO |
| 4 | Discriminant validity—Fornell and Larcker criterion | CLE | 0.692 | | | | | |
| | | CST | 0.523 | 0.628 | | | | |
| | | CON | 0.379 | 0.620 | 0.652 | | | |
| | | CO | 0.409 | 0.553 | 0.566 | 0.661 | | |
| | | QA | 0.440 | 0.553 | 0.566 | 0.404 | 0.646 | |
| | | TO | 0.431 | 0.492 | 0.581 | 0.444 | 0.472 | 0.689 |

Results illustrated in Table 2 indicate that all the values of item reliability and convergent validity of the study's measurement model are above the mentioned cut-off values; it successfully meets the first set of evaluation criteria. Furthermore, the measurement model achieves two discriminant validity criteria through cross-loading and Fornell and

Larcker measures [40]. Thus, it indicates that the assessment of measurement criteria is entirely fulfilled.

### 6.2. Test of Hypotheses

Non-parametric bootstrapping was applied to test the hypothesis by calculating the t-value. Table 3 shows the summary of the path results and the corresponding t-values calculated. For all the paths, a two-tailed *t*-test was used.

**Table 3.** Test of hypotheses.

| Exogenous | Relation with Endogenous | Hypothesis | t-Value | Significant Level (>1.96) |
|---|---|---|---|---|
| CLE | TO | H1: CLE has a significant relationship with TO | 3.403 | Significant |
| CLE | CO | H2: CLE has a significant relationship with CO | 2.988 | Significant |
| CLE | QA | H3: CLE has a significant relationship with QA | 1.158 | Not significant |
| CST | TO | H4: CST has a significant relationship with TO | 1.608 | Not significant |
| CST | CO | H5: CST has a significant relationship with CO | 0.503 | Not significant |
| CST | QA | H6: CST has a significant relationship with QA | 2.916 | Significant |
| CON | TO | H7: CON has a significant relationship with TO | 3.978 | Significant |
| CON | CO | H8: CON has a significant relationship with CO | 2.680 | Significant |
| CON | QA | H9: CON has a significant relationship with QA | 2.161 | Significant |

Results in Table 3 show that the t-value for most of the pathways was above the minimum cut-off level, i.e., 1.96 = 5% [41]. However, for the relations, i.e., client-related causes with quality, consultant-related causes with time overrun, and cost overrun, the t-value is less than 1.96. This means that most of the assumptions are supported and accepted. Thus, there are only three insignificant relations.

### 6.3. Evaluation of Structural Model

The structural model's evaluation assessed the inner model based on two criteria by evaluating the model's predictive capabilities and relationships among constructs. The first criterion is to check the strength of the impact path (β value) of the independent variables to the dependent variables where the cut-off value of β is greater than or equal to 0.1 regardless of its sign (negative or positive) [41]. Table 4 shows the results of the evaluation of the structural model.

**Table 4.** Overall performance of the structural model.

| No | Assessment | Achievement | | | | |
|---|---|---|---|---|---|---|
| 1 | Coefficients of determination, $R^2$ | Outcome: Based on the final model, the $R^2$ values for the structural model are 0.396 for TO, 0.339 for CO, and 0.410 for QA which according to Cohen (1998) specification, the developed model can be classified as having moderate explaining power in representing the impact of the 6 groups of causes and effects on the overall construction project performance | | | | |
| | | Exogenous construct | Endogenous construct | $R^2$ included | $R^2$ excluded | $f^2$ | Interpretation $f^2 \geq 0.02$ (small) $f^2 \geq 0.15$ (medium) $f^2 \geq 0.35$ (large) |
| 2 | Effect size, $f^2$ | CLE | CO | 0.720 | 0.716 | 0.014 | No effect |
| | | | TO | 0.770 | 0.770 | 0.000 | No effect |
| | | | QA | 0.716 | 0.714 | 0.002 | No effect |
| | | CST | CO | 0.720 | 0.714 | 0.021 | Small effect |
| | | | TO | 0.770 | 0.752 | 0.078 | Small effect |
| | | | QA | 0.783 | 0.781 | 0.002 | No effect |
| | | CON | CO | 0.720 | 0.688 | 0.114 | Small effect |
| | | | TO | 0.770 | 0.731 | 0.170 | Medium effect |
| | | | QA | 0.771 | 0.773 | 0.002 | No effect |

**Table 4.** *Cont.*

| No | Assessment | Achievement | | | | | |
|----|------------|-------------|---|---|---|---|---|
| | | Exogenous construct | Endogenous construct | $Q^2$ included | $Q^2$ excluded | $q^2$ | Interpretation $q^2 \geq 0.02$ (small) $q^2 \geq 0.15$ (medium) $q^2 \geq 0.35$ (large) |
| 3 | Predictive relevancy, $q^2$ | CLE | CO | 0.114 | 0.099 | 0.017 | Small relevant |
| | | | TO | 0.160 | 0.149 | 0.013 | Not relevant |
| | | | QA | 0.148 | 0.140 | 0.009 | Not relevant |
| | | CST | CO | 0.114 | 0.118 | −0.005 | Not relevant |
| | | | TO | 0.160 | 0.160 | 0.000 | Not relevant |
| | | | QA | 0.148 | 0.138 | 0.012 | Not relevant |
| | | CON | CO | 0.472 | 0.453 | 0.036 | Small relevance |
| | | | TO | 0.160 | 0.119 | 0.049 | Small relevance |
| | | | QA | 0.148 | 0.125 | 0.027 | Small relevance |

The results illustrated in Table 4 indicate that all the four criteria for assessing the structural model are fulfilled. Hence, this model is statistically validated and can be accepted for further application.

*6.4. Goodness of Fit*

Goodness of fit (GoF) is an index used to identify the geometric mean for endogenous structures of the average community (AVE) and the average determination coefficient ($R^2$) as cited by [33]. The GoF index serves as the basis for validating the global PLS model with a value of 0 to 1. For values of 0.1, 0.25, and 0.36, respectively, the GoF index can be classified into three criteria of small, medium, and high validating capacity [42]. The average AVE for the entire construct variable and the average $R^2$ for all build variables is as shown in Table 5.

**Table 5.** Calculation of goodness of fit.

| Constructs | Average Variance Index (AVE) from Construct Validity and Reliability | $R^2$ Values |
|------------|------------------------------------------------------------------|--------------|
| CLE | 0.612 | |
| CON | 0.769 | |
| CST | 0.694 | |
| TO | 0.582 | 0.396 |
| CO | 0.705 | 0.339 |
| QA | 0.688 | 0.410 |
| Average | 0.675 | 0.381 |

Thus, goodness of fit, $GoF = \sqrt{AVE \times R^2} == \sqrt{0.675 \times 0.381} = 0.507$ (large validating power). The GoF index for this study model was calculated from Table 5 as 0.507. GoF is used to validate the large complex model's prediction power by recording for both measurement and structural parameters [43]. The GoF value of this study shows that the developed model has large validating power. It can be concluded that the empirical data matches the model well and is highly predictive compared to the baseline values.

**7. Discussion on Findings and Benefits**

With the help of the SEM model, this study investigated the issue of change and discovered the cause-and-effect factors of changes. The developed model is appropriate for the implementation phase of the construction cycle and because the respondents were construction practitioners, including clients, consultants, and contractors. It is most beneficial to the contractor's organization in the following ways:

- The contractor can use the model outcome on the rank of change causes as a strategic tool to identify potential causes and reasons. It will accelerate and improve construction performance efficiency if its possible effects are correctly identified and understood.
- The model outcomes will assist the company in selecting the most appropriate change management model. In addition, it will help to manage potential effects, reducing or even eliminating potential problems that could harm project performance as a whole.
- The results can also be used to identify potential project managers with sufficient knowledge and experience in project and change management and appropriate change management tools, models, and techniques.
- The model can also be used to develop high-quality and robust teamwork in managerial positions, allowing them to work under challenging situations caused by unforeseen external factors such as policy changes, economic turmoil, and so on.
- Because the data used to develop the model was current, the results will provide contractors with awareness in updating their understanding of the critical change management approaches in dealing with recent construction's change causes.
- The model results can also be used to inform new company policies aimed at improving construction workers' and engineers' skills to find the best solutions for potential change effects, particularly long-term effects.

## 8. Contribution of the Study

The success of a project is entirely dependent on effective change management. If the adopted change management approach is followed correctly, it will improve work quality and delivery. This study provided a thorough investigation of the causes and effects of the changes in construction. The study's findings are important to the construction industry for the following reasons. Firstly, they are important for identifying the causes aid in avoiding or minimizing the occurrence of change phenomena. Secondly, identifying the effects of change enables practitioners to learn and appreciate the importance of properly executing communication to reduce and mitigate the negative impact on the overall scope and objectives of the project. Finally, the study developed a SEM model to understand the relationships between causes and effects, which will assist practitioners in relating both factors to comprehend the visual aspect of the phenomenon fully. The model can be used as a visual tool to understand the change causes and effects and how to correlate each cause and effect. It also aids in transforming data into decision-making documents that can be used throughout the project lifecycle.

## 9. Conclusions

Change is considered a constant element occurring in any construction project. Changes have a direct impact on the project performance. The practitioners argue that the change usually results in rework, which slow down the work performance and causes monetary loss. This loss is not recoverable and creates an extra burden on the client. Hence, changes must be controlled for efficient performance. Otherwise, change can lead to the failure of the projects. Since changes severely impact project time, cost, and quality, this paper successfully established the relationship between the change causes and effects of UAE construction projects through the PLS-SEM modeling approach. It also presents the development and evaluation of the constructed model to ensure that the model is adequate for the determined relationship representation. The model entails that the contractor (CON) group of causes is considered the most significant cause in originating the change. "Owners' expectations and quality improvement by client" is reported as the major factor in this group. The practitioners pointed out that the owner expectations are very high. Construction activities are labor-intensive and resource-dependent. If any of the staff shows poor workmanship, it affects the work quality. Similarly, if the materials are not uniform in sizes, the quality of the work is affected. It is observed form the model that Quality assurance (QA) is the most significant effect among the effects groups. Poor quality often results in re-

working, which has effect on project time and cost. For the practitioners, it is very essential to manage changes for achieving successful construction projects. However, the scope of the research was limited to the study of the UAE construction industry. However, further investigation should cover other countries and produce a comparative study to understand the issue from different aspects wholly. Based on the study's findings, it is recommended that further investigation be carried out to quantify the effect of change in terms of the time overrun caused, cost overrun, quality, and other relevant quantification measures. Furthermore, other stakeholders such as authorities and other agencies to study their role in generating the issue of change management should be involved in data collection. More research work is required to develop a change management platform that can simplify change through proper approaches, models, and change control methods.

**Author Contributions:** Conceptualization, I.A.R.; and A.E.S.A.A.; methodology, I.A.R.; and A.E.S.A.A.; software, A.E.S.A.A.; A.S.A.M.A.; validation, A.H.M.; I.A.R.; formal analysis, A.E.S.A.A.; investigation, A.E.S.A.A.; resources, A.H.M.; I.A.R.; data curation, A.E.S.A.A.; A.S.A.M.A.; writing—original draft preparation, A.H.M.; N.A.-E.; writing—review and editing, A.H.M.; N.A.-E.; visualization, I.A.R.; supervision, I.A.R.; project administration, I.A.R.; funding acquisition, All the authors jointly. All authors have read and agreed to the published version of the manuscript.

**Funding:** This research received no external funding.

**Institutional Review Board Statement:** The study was conducted according to the guidelines of the Declaration of University Tun Hussein Onn Malaysia as part of doctoral studies.

**Data Availability Statement:** Data was collected from the construction practitioners with commitment that it will be kept confidential and used for academic purpose. This data is used for PhD research project undertaken at University Tun Hussein Malaysia. Data can be made available for editorial purpose as and when required.

**Acknowledgments:** The authors are thankful to Universiti Tun Hussein Onn Malaysia for supporting this research work.

**Conflicts of Interest:** There is no conflict of interest.

## Appendix A

**Table A1.** Cross-Loading Analysis.

| Code | CLE | COS | CON | CO | QA | TO |
|------|-----|-----|-----|-----|-----|-----|
| CLE14 | 0.805 | 0.309 | 0.253 | 0.184 | 0.328 | 0.274 |
| CLE18 | 0.834 | 0.377 | 0.006 | 0.241 | 0.389 | 0.200 |
| COS5 | 0.255 | 0.810 | 0.227 | 0.135 | 0.256 | 0.345 |
| COS7 | 0.439 | 0.890 | 0.227 | 0.223 | 0.375 | 0.378 |
| CON10 | 0.149 | 0.240 | 0.833 | 0.245 | 0.274 | 0.418 |
| CON14 | 0.057 | 0.177 | 0.828 | 0.265 | 0.282 | 0.326 |
| CON16 | 0.166 | 0.232 | 0.786 | 0.331 | 0.230 | 0.287 |
| CO11 | 0.116 | 0.160 | 0.316 | 0.762 | 0.152 | 0.248 |
| CO13 | 0.301 | 0.211 | 0.254 | 0.894 | 0.223 | 0.245 |
| CO15 | 0.201 | 0.149 | 0.263 | 0.760 | 0.206 | 0.184 |
| QA12 | 0.284 | 0.239 | 0.128 | 0.031 | 0.716 | 0.299 |
| QA14 | 0.295 | 0.264 | 0.215 | 0.204 | 0.754 | 0.296 |
| QA8 | 0.414 | 0.352 | 0.353 | 0.273 | 0.848 | 0.461 |
| TO10 | 0.235 | 0.349 | 0.345 | 0.226 | 0.408 | 0.834 |
| TO12 | 0.230 | 0.266 | 0.308 | 0.115 | 0.380 | 0.809 |
| TO6 | 0.221 | 0.387 | 0.355 | 0.305 | 0.337 | 0.748 |

Note: CLE = client-related causes, COS = consultant-related causes, CON = contractor-related causes, CO = cost overrun, QA = quality, TO = time overrun.

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
