# Peer review of "Structural Relationship of Causes and Effects of Construction Changes: Case of UAE Construction"

_sustainability, doi:10.3390/su14020596_

Round 1
Reviewer 1 Report
This manuscript presents an approach to determine the relationship between causes and effects of construction changes, which is an important topic of study among construction researchers. However, given that the causes and effects of construction changes are widely studied topics, the authors have not explicitly justified the need for this study and the use of the structural equation modeling (SEM) technique. Further, the manuscript is not well-drafted, and the authors have not clearly presented the research methodology and result/findings. Therefore, the reviewer is not convinced to consider this manuscript for publication. Following are some specific comments:
- The authors have not justified the need for this study.
- Line 90: What do “conditions” refer to in line 90?
- The author should also explicitly discuss the Partial Least Square Equation Modeling method in the manuscript for readers’ clear understanding of the adopted approach and the evaluation methods. Further, the authors have not stated the approach used for enlisting the changes and effects accounted for in this study (lines 107-109 and Table 1).
- Table 2: It would be more informative to present these tabulated values in graphical form (e.g., pie chart).
- Figure 2: What do the six latent variables mean?
- Table 3: The assessment methods presented in Table 3 are not discussed in the manuscript.
- Table 4: Indicate p-values for all t-tests.
- A discussion on results and findings is not presented in the manuscript.
- The authors should discuss the contribution of this study.
Reviewer 2 Report
The article evaluates the projects in UAE to understand the causes (due to contractor, client, consultant) and their effects (cost, time, quality) in construction industry with use of structural equation modelling approach.
- Line 17-18: In following sentence, please elaborate changes "this paper addresses the issues occurring in construction projects due to changes."
- Typos: line 32, 40 140-151 etc.
- Improve keywords.
- Rewrite 60-62, 74-76, 176-177, Table 3 No. 1 & 3, also improve discussion on goodness-of-fit
- Line 48: Ref [6], please use authors name.
- I think it would be appropriate to discuss the salient features of construction industry of UAE, workforce demographics, types of projects, contracts etc.
- In abstract software is given, however there is no mention of software in the remaining paper.
- It is also not clear what types of questions were asked from respondents on 5-Likert scale.
- That are the limitations of current study and recommendations for future research?
- There is 1 ref from 2019, 2 from 2020 and no ref from 2021. I think a more critical review of the state of the art research is required to establish need of the study.
- The same is reflected in description of methods and interpretation of results. Authors are advised to improve discussions.
Reviewer 3 Report
Dear autors,
The paper is very well done, only some paragraphs need some revisions to the English language (especially the abstract).
Another advice would be to expand your conclusions a bit for example in case of the GoF index for this study model.

Round 2
Reviewer 2 Report
Line 27 was collected, also a construction project.
Line 43 past sentence?
Line 48 are have given?
Line 58 use negative effects instead negative situation
Remove the last sentence from line 63 to 65 or fit in context of your research.
67 considered to be
Line 68 seem repetitive, you may remove it or reword.
Line 75 : You may remove the following part of the sentence "change can occur in various ways, such as".
Line 96-98: Starting of sentence should be different for readability.
Line 101-102: Sentence seem repetitive, you may modify it.
Lin3 77-99 excessive use of Change.
Line 149-161 unbold
Figure 1 has disturbed, also there is issue of spacing of captions after figure and table numbers.
Check line 257 to 258
